# Responses of soil labile organic carbon stocks and the carbon pool management index to different vegetation restoration types in the Danxia landform region of southwest China

Ye Xiao[1], Yuguang Zhang[2], Jiumei Long[1], Kang Luo[3], Zhigang Huang[4]*

1 Department of Resources and Environment, Zunyi Normal College, Zunyi, China, 2 Ecology and Nature Conservation Institute, Chinese Academy of Forestry, Beijing, China, 3 Guizhou Xishui Subtropical Evergreen Broad-leaved Forest National Nature Reserve Management Bureauin, Guizhou Xishui, China, 4 Department of Biology and Agricultural Science and Technology, Zunyi Normal College, Zunyi, China

* huangzhigang2016@sina.com

## Abstract

Soil organic carbon (SOC) is an important index for evaluating soil quality in the process of ecological restoration. It plays an important role in increasing soil carbon storage, improving soil texture, and promoting plant growth. Nevertheless, dating the variation in SOC and labile SOC fractions (LOCFs) during ecological restoration processes has not been sufficiently elucidated. To enrich our comprehension of the responses of SOC and its labile fractions to different vegetation restoration types, five vegetation restoration types were selected in the Danxia landform region of southwest China, namely, shrub (SH), bamboo forest (BF), Chinese fir forest (CFF), evergreen broad-leaved forest (EBF), and mixed coniferous-broadleaf forest (MCBF). The concentrations and stocks of SOC and LOCFs, such as dissolved organic carbon (DOC), microbial biomass organic carbon (MBC), and easily oxidizable organic carbon (EOC), and the carbon pool management index (CPMI) were investigated. Results showed that the different vegetation restoration types significantly influenced SOC stocks ($P < 0.05$), and the concentrations and stocks of SOC and LOCFs decreased with increasing soil depth in different vegetation types, except for MBC in BF and CFF. Additionally, BF and EBF had significantly higher total SOC stocks (92.75 t ha⁻¹ and 60.13 t ha⁻¹) compared with the three other vegetation types (26.18–47.48 t ha⁻¹) at a depth of 0–30 cm. The largest total DOC stock was observed in EBF, while BF and EBF had significantly higher MBC and EOC stocks than SH, CFF, and MCBF ($P < 0.05$). Compared with SH, the CPMI increased by 49.7%, 32.9%, and 35.2% in BF, CFF, and EBF, respectively, except for the MCBF. SOC and LOCFs were closely related to soil physicochemical properties, and total nitrogen, total phosphorus and moisture had a pronounced effect on them. However, higher SOC and LOCFs stocks, and CPMI were observed in BF and EBF than in the other vegetation types. This result suggests that the two plant types exhibited better ability to sequester carbon than the other vegetation types. Overall, vegetation restoration promoted the

**Data availability statement:** All relevant data are within the manuscript and its Supporting Information files.

**Funding:** Guizhou Provincial Basic Research Program (Natural Science) (Qiankehejichu-ZK[2024]yiban 674 and 691) and Scientific Research Project of Zunyi Normal University (Zunshi BS [2019]30). Our funding is funded by government agencies. Funding agencies provide financial support for our research design, data collection and analysis, publication decisions or manuscript preparation.

accumulation of both SOC and its fractions, the results of which varied among the different vegetation types.

## Introduction

Danxia landform is a unique landform formed on red beds, distinguished by steep slopes [1]. It is found primarily in China, the Western United States, Central Europe and Australia; however, it is most widespread in China. China's Danxia is a sedimentary red land landscape formed by the combination of crustal uplift, weathering, and water erosion [2]. It is largely distributed in China's tropical and subtropical humid, temperate humid and semi-humid, and semi-arid and arid regions, and the Tibetan Plateau [3]. The red beds for Danxia's development are mostly red continental clastic rocks, particularly coarse clastic rocks, with conglomerates and sandstones as the major ones [4]. Consequently, ions and nutrients in Chinese Danxia soils are easily lost under the long-term influence of external forces, resulting in acidic to highly acidic soil with a loose physical structure, thin soil layer, and exposed parent material [5]. Affected by its peculiar geological structure, humid climate, and human activities, the ecological environment of the Danxia landform area is extremely fragile and gradually deteriorating, seriously threatening the local environment and social economy [6]. However, Danxia landforms are relatively under-researched compared with other rock geomorphologies, such as karst and granite landforms. Moreover, its ecological and associated problems have not received sufficient attention.

Vegetation restoration is a highly effective practice for controlling desertification and restoring degraded ecosystems [7,8]. This practice cannot only improve soil texture, increase biodiversity, and reduce soil erosion, but it also increases the stock of soil carbon pools. As a driving force of ecosystem functions, the soil organic carbon (SOC) pool plays a critical role in regulating soil fertility, moisture retention, enhancing soil aggregate structure, and mitigating atmospheric $CO_2$ levels [9]. Thus, SOC is an important indicator that reflects soil quality, and it is used to evaluate the restoration effect on degraded ecosystems, which is one of the key factors for carbon cycling research in terrestrial ecosystems [10,11]. The decline in SOC content will not only lead to a reduction in soil fertility and water-holding capacity but also affects greenhouse gas emissions. Therefore, protecting and sequestering SOC are critical for slowing down the global greenhouse effect [12,13]. Compared with SOC, labile SOC fractions (LOCFs), such as dissolved organic carbon (DOC), microbial biomass carbon (MBC) and easily oxidized organic carbon (EOC), comprise only a fraction of total organic carbon (TOC) in soil, but respond to land use more rapidly due to their poor stability, ease of movement, susceptibility to decomposition by microorganisms, and rapid turnover [14]. In addition, they can affect the physical and chemical properties of soil and the uptake and utilization process by plants, altering soil structure and soil fertility, and indirectly affecting soil carbon cycle. Therefore, LOCFs cannot only help reveal dynamic changes in SOC but also sensitively reflect slight changes in the soil carbon pool resulting from soil management, plant communities, and environmental changes. Knowledge about these fractions in any land-use system will help determine soil quality and sustainability. Many studies have explored the dynamics of total SOC and LOCFs after vegetation restoration in ecologically fragile areas, such as deserts [15], karst [16], loess plateaus [17], and severely eroded red soil regions [18], and then evaluated the effectiveness of vegetation restoration. Blair et al. (1995) [19] first proposed the concept of the soil carbon pool management index (CPMI), which is a comprehensive assessment of soil quality from the two aspects, i.e., soil TOC and LOCFs. It can reflect the degradation or renewal condition of soil quality under different management actions. An increase in soil

CPMI indicates that the management mode exerts the effect of increasing soil fertilization, and the soil structure will develop into a benign level; conversely, soil quality will become malignant [20]. Moreover, various indices have been developed to assess the sustainability of land use practices, such as the lability index (LI) and the carbon management index (CMI).

Chishui Danxia is the largest and most spectacular Danxia landform in China, encompassing an area of approximately $5.19 \times 10^4$ hm² [21], and located in southwestern China. In August 2010, it officially entered the World Heritage List with five other sites to form "China Danxia" [3]. Before the 1990s, to solve the conflict between the increasing population and the decreasing amount of land in this region, people had to destroy forests and cultivate crops on steep slopes, resulting in resource scarcity, low forest cover, and severe soil erosion. These conditions pose a serious threat to the ecological environment. The remarkable success of natural succession and reforestation as effective strategies for curbing soil erosion and other detrimental impacts associated with land degradation, particularly in ecological vulnerable zones. Therefore, with the support of the national "Grain for Green" program in the 1990s, the local government also took positive and effective measures to enhance vegetation coverage rate, such as afforestation, returning farmlands into forests, and forest conservation. Since then, the ecological environment of Chishui Danxia district has been gradually recovering, with vegetation coverage rate increasing from 55.4% in 2000 to 67.4% in 2016 [22], and currently over 85%, successfully achieving the goal of the "red-layer oasis". Many studies have documented the influences of vegetation restoration and land utilization on SOC and its fractions in the karst areas of southwest China [23–25]. As for the Danxia landform area, people's research mainly focuses on the vegetation, biodiversity, geological structure and tourism development [3,26,27]. However, relatively fewer investigations have been conducted to evaluate the ecological restoration effect of Danxia landform in this region from the perspective of SOC.

In this study, our primary objectives were as follows: (a) to determine how the vegetation types affect the changes of soil carbon fractions and whether the effect is species-specific; (b) to find out how do the LI and CPMI of carbon change in various vegetation types; and (c) to ascertain which vegetation types are optimal for restoring deteriorated land, specifically in the western region of Guizhou. Our research results provide basic data for the ecological benefit evaluation of different vegetation restoration techniques in China Danxia landform area.

## Materials and methods

### Study site

The study was conducted at the Sancha River Experimental Station of the Xishui Subtropical Evergreen Broad-leaved Forest National Nature Reserve (28°07′–28°34′N, 105°50′–106°29′) in the Chishui Danxia landform area, Guizhou Province, southwest China. The region experiences mid-subtropical to warm temperate humid monsoon climate, with an average annual precipitation of 770.3–1661 mm and temperature of 14.7 °C. Its geology is characterized by the Danxia landform, which is composed of Cretaceous continental clastic red beds and presents a "V"-shaped valley topography with a relative elevation difference of 400–600 meters [28]. The majority of the soil is acidic and predominantly purple, followed by yellow soil and then minimal yellow-brown soil.

The reserve is a typical Danxia landscape, the region experienced a sharp decline in forest cover and severe soil erosion as a result of extensive deforestation, overgrazing, reclamation of steep slopes, and engineering construction before 1990s. Supported by the policy of returning farmlands into forests since the 1990s, the local administration has diligently augmented the supervision and conservation of this region through forest closure. The diminished mountain

forest has gradually regenerated and the vegetation density has noticeably increased through natural succession and reforestation.

## Sample collection and chemical analyses

Our research obtained the forestry administrative license of Xishui National Nature Reserve Administration of Guizhou Province. In May 2022, five types of restored vegetation were investigated in this study: shrub (SH), mixed conifer–broadleaf forest (MCBF), evergreen broad-leaved forest (EBF), Chinese fir forest (CFF), and bamboo forest (BF). Three representative 20 m × 20 m plots were randomly established in a sample field for each vegetation type. Table 1 provides basic information about the five vegetation types. To mitigate potential edge effects, the surveyed plots of identical vegetation type were separated by exceeding 10 m from the forest boundary. Five sampling sites were set up along a "Z"-shaped area in each plot, and soil samples were collected from the 0–10, 10–20, and 20–30 cm soil layers by using a soil metal sampler (internal diameter: 4 cm) after removing the litter layer. The soil samples from the same soil layer at the five sampling sites in the same plot were then mixed together to form a composite sample. Finally, forty-five composite soil samples were obtained from five vegetation types (five vegetation types × three soil depths × three plots). Moreover, three soil cores were gathered from each soil layer in each plot to ascertain bulk density (BD). Each sample was carefully enveloped in sterile, airtight plastic bag and promptly preserved in a portable cryogenic refrigerator until their relocation to the laboratory. The soil samples were divided into two sub-samples. One portion of fresh soil samples was stored at 4°C to measure soil moisture, MBC, and DOC within 1 week after sampling. The remaining portion of the samples was naturally air-dried, ground, and sieved for the analysis of SOC, other physicochemical properties (through a 2 mm mesh) and EOC (through a 0.15 mm mesh). The remaining gravel content (>2mm) was also weighted. The basic soil properties are presented in Table 2.

Soil physical and chemical properties: bulk density (BD) was measured via the ring knife method, and soil pH was determined in soil suspension (soil: water = 1:2.5, w/v) by using a glass electrode pH meter. Soil moisture content was dried to constant weight at 105°C. Soil total nitrogen (TN) and total phosphorus (TP) were detected via the automatic Kjeldahl nitrogen determination method and the perchloric acid–hydrofluoric acid digestion procedure method, respectively [29].

SOC was determined using the potassium dichromate oxidation–reduction method [29].

The analyses of LOCFs were as follows. Soil DOC was analyzed according to the method described by Ghani et al. [30]. Soil MBC was calculated by the chloroform fumigation–potassium sulfate leaching method [31]. After 40 g of fresh soil samples were fumigated with chloroform vapor for 24 h in a vacuum desiccator, and the fumigated and unfumigated soil samples were extracted with 0.5 mol·L$^{-1}$ K$_2$SO$_4$ solution, the conversion coefficient of 0.38

**Table 1. Basic situation of different vegetation types.**

| Vegetation types | Elevation (m) | Slope (°) | Aspect | Soil type | Dominant tree species |
|---|---|---|---|---|---|
| Shrub (SH) | 928 | 24 | West by north 26° | Yellow soil | *Coriariasinica, Vitex negundo, Zanthoxylum planispinum, Lonicera ligustrina* |
| Evergreen broad-leaved forest (EBF) | 991 | 32 | West by north 73° | Yellow soil | *Lithocarpus glabra, Phoebe zhennan, Clethra pinfaensis* |
| Chinese fir forest (CFF) | 1025 | 41 | West by north 77° | Yellow soil | *Cunninghamia lanceolata* |
| Mixed conifer -broadleaf forest (MCBF) | 955 | 20 | East by south 63° | Yellow soil | *Fokieniahodginsis, Pinus massoniana,Castanopsischunii, Elaeocarpus japonicus* |
| Bamboo forest (BF) | 1028 | 11 | West by north 9° | Yellow soil | *Phyllostachys pubescens* |

**Table 2. Mean (±SE, n = 3) soil basic physical and chemical properties in the five vegetation types.**

| Vegetation types | Soil layers | pH | Moisture (%) | BD (g·cm⁻³) | TN (mg·kg⁻¹) | TP (g·kg⁻¹) |
|---|---|---|---|---|---|---|
| SH | 0–10 | 7.62±0.11Ab | 21.73±1.51 Ba | 1.75±0.07 Aa | 1.36±0.21Ba | 0.42±0.00 Ba |
| | 10–20 | 8.18±0.04 Aa | 16.59±0.67 Ab | 1.86±0.18 Aa | 0.69±0.06 Bb | 0.41±0.01b Ba |
| | 20–30 | 8.11±0.04 Aa | 14.51±1.77 Bb | 1.95±0.10 Aa | 0.61±0.01 Bb | 0.39±0.00b Ab |
| | Average | 7.97±0.06 A | 17.61±1.32 C | 1.87±0.10 AB | 0.89±0.09 BC | 0.41±0.00 B |
| BF | 0–10 | 5.49±0.09 Bb | 29.50±1.96 ABa | 1.29±0.04 Bb | 3.32±0.25 Aa | 0.76±0.13 Aa |
| | 10–20 | 5.91±0.09 Ba | 22.08± 1.45 Ab | 1.50±0.08Aab | 1.75±0.13 Ab | 0.62±0.05 Aa |
| | 20–30 | 5.88 ±0.00 Ba | 21.00±0.67 Ab | 1.64±0.21 Aa | 1.55±0.34 Ab | 0.52±0.09 Aa |
| | Average | 5.76±0.06 B | 24.19±1.36 AB | 1.48±0.11 C | 2.21±0.24 A | 0.63±0.10 A |
| CFF | 0–10 | 5.65±0.61 Ba | 27.13±5.77 ABa | 1.53±0.06 ABa | 1.45±0.61 Ba | 0.26±0.01 Ca |
| | 10–20 | 5.35±0.57 BCa | 22.18±6.28 Aa | 1.61±0.07 Aa | 0.79±0.18 Ba | 0.24±0.05 Ca |
| | 20–30 | 5.33±0.51 BCa | 18.08±3.00 ABa | 1.76±0.08 Aa | 0.69±0.11 Ba | 0.22±0.03 Ba |
| | Average | 5.44±0.57 B | 22.46±5.02 B | 1.63±0.07 BC | 0.98±0.30 B | 0.24±0.03 C |
| MCBF | 0–10 | 4.84±0.39 BCa | 21.73±3.75 Ba | 1.76±0.20 Aa | 0.89±0.12 Ba | 0.21±0.01 Ca |
| | 10–20 | 4.98±0.35 Ca | 16.05±1.52 Ab | 1.81±0.26 Aa | 0.36±0.04 Cb | 0.17±0.02 Cb |
| | 20–30 | 4.91±0.29 Ca | 14.51±0.52 Bb | 1.88±0.06 Aa | 0.27±0.03 Bb | 0.14±0.01 Bb |
| | Average | 4.91±0.34 C | 17.43±1.93 C | 1.82±0.17 A | 0.51±0.06 C | 0.17±0.01 C |
| EBF | 0–10 | 4.51±0.15 Cb | 35.53±2.98 Aa | 1.51±0.14 ABb | 2.03±0.68 Ba | 0.25±0.05 Ca |
| | 10–20 | 5.03±0.14 Ca | 23.73±1.37 Ab | 1.80±0.17 Aab | 0.92±0.17 Bb | 0.21±0.02 Ca |
| | 20–30 | 5.21±0.22BCa | 23.41±3.47 Ab | 2.04±0.09 Aa | 0.74±0.18 Bb | 0.20±0.01 Ba |
| | Average | 4.92±0.17 C | 27.56±2.61 A | 1.78±0.14 AB | 1.23±0.34 B | 0.22±0.03 C |

Note:SH: shrub, BF: bamboo forest, CFF: Chinese fir forest, MCBF: mixed conifer–broadleaf forest, and EBF: evergreen broad-leaved forest. Uppercase letters (A, B and C) indicate significant differences among the different vegetation types in the same soil layer ($P < 0.05$); Lowercase letters (a and b) indicate significant difference among the same vegetation types in different soil layers ($P < 0.05$). ± indicates standard error of mean.

was utilized in the compulation of MBC. EOC was detected via oxidation with 0.333 mol L⁻¹ KMnO₄ [19].

## Determination of soil carbon stocks and CPMI

The total carbon stock can be calculated as follows [32]:

$$Total\ C\ stock = \frac{\sum_{i=1}^{n} SOCC_i \times BD_i \times H_i \times (1 - \partial_i)}{10} \tag{1}$$

where, $SOCC_i$, $BD_i$, and $H_i$ are the SOC concentrations (g·kg⁻¹), the bulk density (Mg m⁻³), and the thickness (cm), respectively; $\delta_i$ is the proportion (%) of course (>2 mm) fragments; i represents a specific soil layer; and n is the total number of soil layers. SH soil was used as reference soil to identify the effects of different vegetation types on the variations of SOC and LOCFs. According to previous studies [33,34], the soil CPMI values of each vegetation type were calculated according to the following formulas:

$$Carbon\ pool\ index\ (CPI) = \frac{SOC\ of\ the\ sample\ (g \cdot kg^{-1})}{SOC\ of\ the\ reference\ sample\ (g \cdot kg^{-1})} \tag{2}$$

$$Lability\ of\ Carbon\ (L) = \frac{EOC\ (g \cdot kg^{-1})}{SOC(g \cdot kg^{-1}) - EOC\ (g \cdot kg^{-1})}, \tag{3}$$

$$Lability\ index\ \left(LI\right) = \frac{Lability\ of\ C\ in\ each\ sample\ soil}{Lability\ of\ C\ in\ the\ reference\ soil} \tag{4}$$

$$Carbon\ pool\ management\ index\left(CPMI\right) = CPI \times LI \times 100 \tag{5}$$

## Statistical analysis

All data were statistically analyzed using Excel 2019 and SPSS 21.0 (SPSS Inc., USA), and checked residual normality. A two-way analysis of variance (ANOVA) was conducted to investigate the impact of vegetation types and soil layers on SOC, its labile fractions and the other indices. One-way ANOVA were initially served to evaluate potential variations on various indices among the five vegetation types within the same soil layer, subsequently extending to compare the differences between three soil layers of identical vegetation type. The least significant differences ($P < 0.05$) was deployed to compare the differences among means. To evaluate the effects of soil factors on SOC and its LOCFs, redundancy analysis (RDA) and mapping was conducted on the online tool of Majorbio Cloud Platform (https://cloud.Majorbio.com/page/tools/). Figures were generated using MS Excel 2019.

## Results

### Changes in SOC concentrations and stocks

SOC concentrations were notably influedced by vegetation restoration types and soil layers (Table 3), demonstrating a tendency to decline with increasing soil depth across all the vegetation types (Fig 1a). Within the three soil layers, BF had significantly higher SOC concentrations than the four other vegetation types (Fig 1a). Overall, the mean SOC concentrations in the different vegetation types ranged from 4.93-22.09 g kg$^{-1}$ at a depth of 0–30 cm and followed the order of BF> EBF> CFF> SH> MCBF. Moreover, SOC levels within BF and EBF exhibited notably higher values than those observed in the other vegetation types.

A decreasing trend in SOC stocks with depth was also appeared among the five vegetation types (Fig 1b). Furthermore, the contribution of SOC stocks in the 0–10 cm soil layer of the five vegetation types varied from 37.2% to 66.8% and was notably higher than those of the two deep soil layers. Similar to SOC concentration, significantly higher SOC stocks were found in BF across the three soil depths compared with the other vegetation types. At a depth of 0–30 cm, total SOC stocks in BF, EBF, CFF, SH, and MCBF were 92.75, 60.13, 47.48, 46.56, and 26.18 t ha$^{-1}$, respectively. These findings showed that BF and EBF had significantly higher mean SOC concentrations and total SOC stocks than SH, CFF, and MCBF.

**Table 3. Two-way ANOVA of the effects of different vegetation types (VT) and soil depths (SD) on SOC and LOCFs.**

| Factors | SOC (g kg$^{-1}$) | | DOC (mg kg$^{-1}$) | | MBC (mg kg$^{-1}$) | | EOC (mg kg$^{-1}$) | |
|---|---|---|---|---|---|---|---|---|
| | F | P | F | P | F | P | F | P |
| VT | 181.82 | <0.001 | 53.63 | <0.001 | 187.01 | <0.001 | 72.51 | <0.001 |
| SD | 235.12 | <0.001 | 311.04 | <0.001 | 165.17 | <0.001 | 245.52 | <0.001 |
| VT×SD | 14.62 | <0.001 | 8.80 | <0.001 | 50.76 | <0.001 | 9.94 | <0.001 |

SOC: soil organic carbon, DOC: dissolved organic carbon, MBC: microbial biomass carbon and EOC: easily oxidized organic carbon.

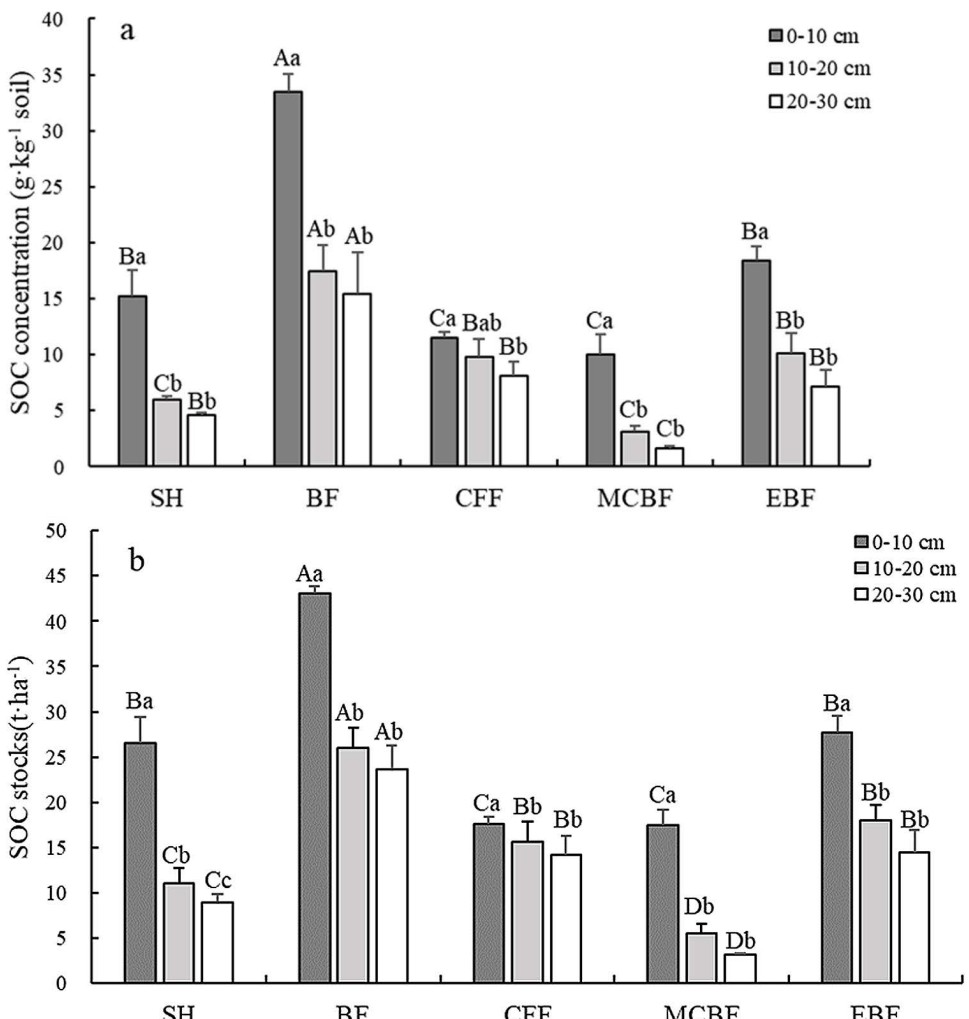

**Fig 1. SOC concentration (a) and stocks (b) in each soil layer of the different vegetation types.** (SOC: soil organic carbon. SH: shrub, BF: bamboo forest, CFF: Chinese fir forest, MCBF: mixed conifer–broadleaf forest, and EBF: evergreen broad-leaved forest. Uppercase letters (A, B, C and D) indicate significant differences among the different vegetation types in the same soil layer ($P < 0.05$); Lowercase letters (a, b and c) indicate significant differences among the same vegetation types in different soil layers ($P < 0.05$). Error bars represent standard error.).

## Variations in soil LOCFs concentrations and stocks

The concentrations and stocks of LOCFs (DOC, MBC, and EOC) varied across vegetation types and soil depths. Fig 2 and Table 4 respectively present the concentrations of LOCFs and their ratios to SOC in different vegetation restoration types. Overall, the concentrations of soil DOC, MBC (excluding BF and CFF), and EOC further exhibited a declining trend with increasing soil depth for all the vegetation types (Figs 2a, 2b and 2c).

Within the same soil layer, soil DOC concentrations varied significantly among vegetation types (Fig 2a). At a depth of 0–30 cm, the mean soil DOC concentrations of BF and EBF were 90.60 mg kg$^{-1}$ and 84.90 mg kg$^{-1}$, respectively, which were significantly higher than those of the three other vegetation types (55.45–65.88 mg kg$^{-1}$) ($P < 0.05$). For the five vegetation restoration types, DOC comprised extremely small proportions (0.39%–1.64%) of SOC in the three soil layers. MCBF held a significantly higher average DOC:SOC (1.38%) than the four other vegetation types (0.40%–0.92%).

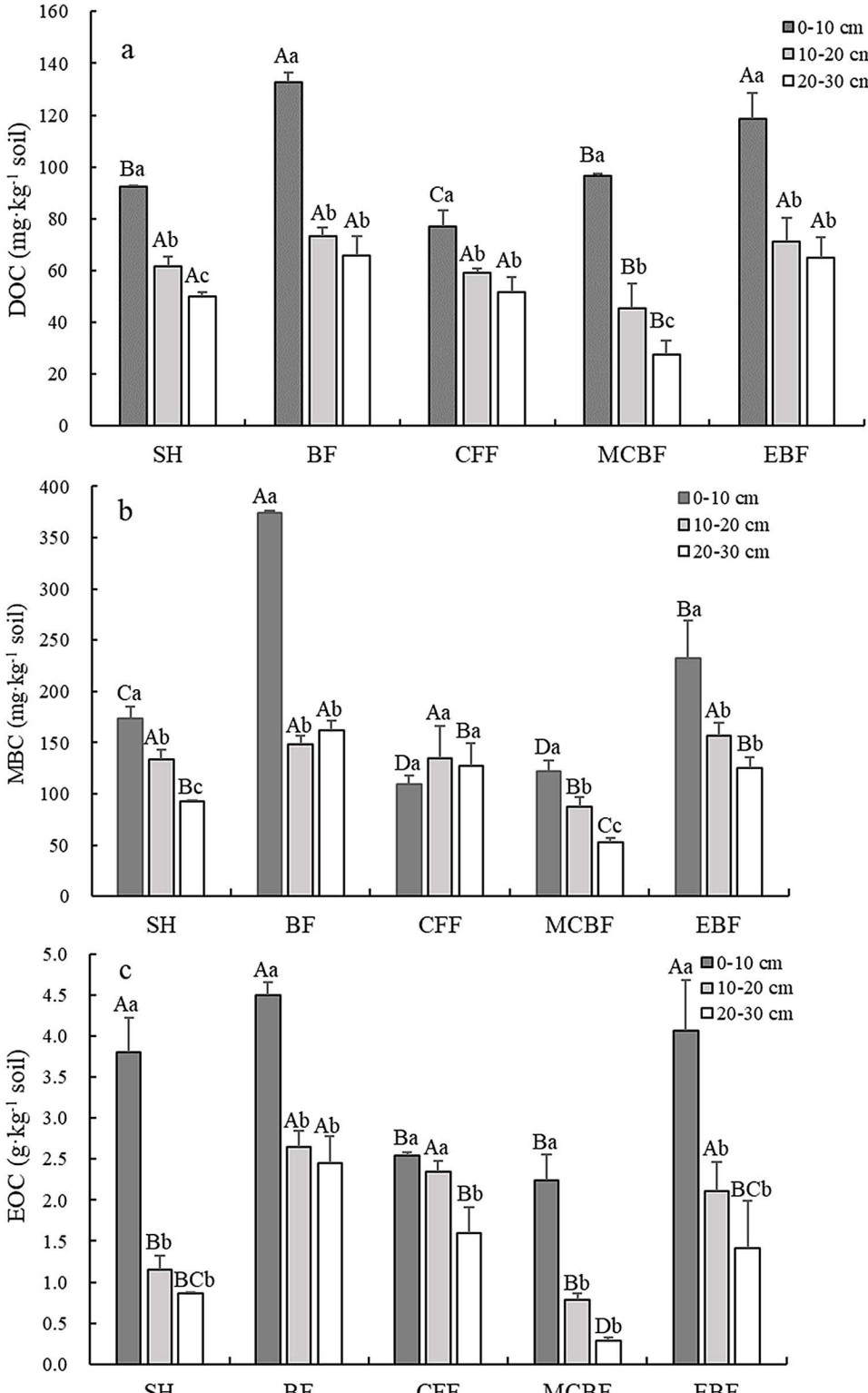

**Fig 2. LOCF concentrations of the different vegetation types.** a) DOC, b) MBC, and c) EOC. (DOC: dissolved organic carbon, MBC: microbial biomass carbon and EOC: easily oxidized organic carbon. SH: shrub, BF: bamboo forest, CFF: Chinese fir forest, MCBF: mixed conifer–broadleaf forest, and EBF: evergreen broad-leaved forest. Uppercase letters (A, B, C and D) indicate significant differences among the different vegetation types in the same soil layer ($P < 0.05$); Lowercase letters (a, b and c) indicate significant differences among the same vegetation types in different soil layers ($P < 0.05$). Error bars represent standard error.).

**Table 4. The proportion of LOCFs (DOC, MBC, and EOC) in total SOC (%) of the five vegetation types.**

| Soil layer | SH | BF | CFF | MCBF | EBF |
|---|---|---|---|---|---|
| DOC:SOC | | | | | |
| 0–10 cm | 0.61 ± 0.10 Bb | 0.40 ± 0.01 Ca | 0.67 ± 0.08 Ba | 0.98 ± 0.16 Aa | 0.65 ± 0.07 Ba |
| 10–20 cm | 1.04 ± 0.01 ABa | 0.43 ± 0.04 Ca | 0.62 ± 0.09 BCa | 1.53 ± 0.49 Aa | 0.71 ± 0.05 BCa |
| 20–30 cm | 1.10 ± 0.09 Ba | 0.39 ± 0.04 Da | 0.65 ± 0.04 Ca | 1.64 ± 0.19 Aa | 0.94 ± 0.14 BCa |
| Average | 0.92 ± 0.07 B | 0.40 ± 0.03 D | 0.64 ± 0.07 C | 1.38 ± 0.28 A | 0.76 ± 0.09 C |
| MBC:SOC | | | | | |
| 0–10 cm | 1.14 ± 0.09 Ac | 1.12 ± 0.06 Aa | 0.95 ± 0.08 Aa | 1.24 ± 0.21 Ab | 1.27 ± 0.21 Aa |
| 10–20 cm | 2.25 ± 0.03 Ba | 0.86 ± 0.17 Da | 1.37 ± 0.15 Ca | 2.88 ± 0.35 Aa | 1.57 ± 0.17 Ca |
| 20–30 cm | 2.04 ± 0.08 Bb | 0.96 ± 0.16 Ca | 1.62 ± 0.45 Ba | 3.19 ± 0.31 Aa | 1.79 ± 0.30 Ba |
| Average | 1.81 ± 0.07 B | 0.98 ± 0.13 D | 1.31 ± 0.22 C | 2.43 ± 0.29 A | 1.54 ± 0.23 BC |
| EOC:SOC | | | | | |
| 0–10 cm | 25.05 ± 1.02 Aa | 13.45 ± 0.18 Ba | 22.00 ± 0.84 Aa | 22.39 ± 2.29 Aa | 22.03 ± 1.80 Aa |
| 10–20 cm | 19.26 ± 1.93 Ab | 15.33 ± 1.06 Aa | 24.38 ± 3.37 Aa | 26.51 ± 8.45 Aa | 20.97 ± 1.76 Aa |
| 20–30 cm | 18.99 ± 1.47 Ab | 14.28 ± 1.27 Aa | 19.75 ± 0.66 Aa | 16.73 ± 3.27 Aa | 20.50 ± 4.80 Aa |
| Average | 21.10 ± 1.47 A | 14.35 ± 0.84 B | 22.04 ± 1.63 A | 21.88 ± 4.67 A | 21.17 ± 2.79 A |

Note: ± indicates standard error of mean. Uppercase letters (A,B,C and D) indicate significant differences among the different vegetation types in the same soil layer ($P < 0.05$); Lowercase letters (a,b and c) indicate significant differences among the same vegetation types in different soil layers ($P < 0.05$). SOC: soil organic carbon, DOC: dissolved organic carbon, MBC: microbial biomass carbon and EOC: easily oxidized organic carbon. SH: shrub, BF: bamboo forest, CFF: Chinese fir forest, MCBF: mixed conifer–broadleaf forest, and EBF: evergreen broad-leaved forest.

However, total DOC stocks ranged from 0.29 t·ha$^{-1}$ to 0.43 t·ha$^{-1}$ in the five vegetation types at a depth of 0–30 cm. The largest total DOC stock was recorded in EBF, significantly higher than that of CFF and MCBF, but not significantly different from those of SH and BF (Table 5).

Fig 2b revealed that in all the vegetation types, soil MBC concentrations showed an inconsistent fluctuating trend in the soil vertical profile, and only CFF demonstrated no significant differences among the three soil layers. The was a notable distinction in MBC values within the identical soil layer across varied vegetation types ($P < 0.05$). At a depth of 0–30 cm, the mean MBC concentrations in BF and EBF were 228.17 mg kg$^{-1}$ and 171.42 mg kg$^{-1}$, respectively, which were significantly higher than those in the three other types (87.36–133.02 mg kg$^{-1}$). MBC accounted for 0.86%–3.19% of SOC, and the average proportion in MCBF was 2.43%, which was significantly higher than those of the other vegetation types (0.98%–1.81%) (Table 4). Overall, significantly higher total MBC stocks were recorded in BF and EBF with 0.96 t·ha$^{-1}$ and 0.88 t·ha$^{-1}$, respectively, compared with the three other vegetation types with 0.50–0.73 t·ha$^{-1}$ at a depth of 0–30 cm (Table 5).

Significant differences in EOC concentrations were also observed among the different vegetation types in the same soil layer (Fig 2c). In general, BF and EBF exhibited significantly higher mean EOC concentrations (3.2 g kg$^{-1}$ and 2.53 g kg$^{-1}$, respectively) than the three other vegetation types (1.10–2.16 g kg$^{-1}$) at a soil depth of 0–30 cm. Compared with DOC and MBC, EOC accounted for the highest proportion (13.45%–26.51%) of SOC (Table 4). BF had a mean EOC:SOC of 14.35%, significantly lower than those of the four other vegetation types (21.10%–22.04%). The highest total stock of EOC was found in BF (13.73 t·ha$^{-1}$), followed by that in EBF (12.77 t·ha$^{-1}$), which were significantly higher than those in the three other vegetation types (5.85–10.48 t·ha$^{-1}$) (Table 5).

## Changes of CPMI in the different vegetation types

For the four vegetation types, no consistent trend was noted in the values of lability of C (L), lability index (LI), carbon pool index (CPI), and carbon pool management index (CPMI) with

**Table 5. LOCFs (DOC, MBC, and EOC) stocks in the different vegetation types.**

| Soil layer | SH | BF | CFF | MCBF | EBF |
|---|---|---|---|---|---|
| DOC (t·ha⁻¹) | | | | | |
| 0–10 cm | 0.16±0.01 ABa | 0.17±0.03 Aa | 0.12±0.02 Ba | 0.18±0.03 Aa | 0.18±0.01 Aa |
| 10–20 cm | 0.11±0.02 ABb | 0.11±0.01 ABb | 0.09±0.01 Bb | 0.09±0.02 Bb | 0.13±0.01 Ab |
| 20–30 cm | 0.10±0.00 Bb | 0.10±0.01 Bc | 0.08±0.01 Bb | 0.05±0.01 Cb | 0.12±0.02 Ab |
| 0–30 cm | 0.37±0.03 AB | 0.38±0.05 AB | 0.29±0.04 C | 0.32±0.06 BC | 0.43±0.04 A |
| MBC (t·ha⁻¹) | | | | | |
| 0–10 cm | 0.30±0.01 Ba | 0.48±0.02 Aa | 0.17±0.02 Ca | 0.23±0.03 Ca | 0.35±0.03 Ba |
| 10–20 cm | 0.25±0.04 ABa | 0.22±0.02 ABb | 0.21±0.04 ABa | 0.17±0.04 Bab | 0.28±0.01 Ab |
| 20–30 cm | 0.18±0.01 Bb | 0.26±0.02 Ab | 0.20±0.03 Ba | 0.10±0.01 Cb | 0.25±0.02 Ab |
| 0–30 cm | 0.73±0.06 B | 0.96±0.06 A | 0.58±0.09 C | 0.50±0.08 C | 0.88±0.06 A |
| EOC (t·ha⁻¹) | | | | | |
| 0–10 cm | 6.64±0.46Aa | 5.80±0.03 Aa | 4.07±0.34 Ba | 3.86±0.11 Ba | 6.10±0.68 Aa |
| 10–20 cm | 2.16±0.54 Bb | 3.97±0.07 Ab | 3.61±0.13 Aa | 1.47±0.14 Bb | 3.78±0.51 Ab |
| 20–30 cm | 1.68±0.05 Cb | 3.96±0.05 Ab | 2.51±0.12 Bb | 0.52±0.11 Dc | 2.89±0.17 Bb |
| 0–30 cm | 10.48±1.05 C | 13.73±0.14 A | 10.19±0.59 C | 5.85±0.36 D | 12.77±1.36 B |

Note: ± indicates standard error of mean. Uppercase letters (A, B and C) indicate significant differences among the different vegetation types in the same soil layer (*P < 0.05*); Lowercase letters (a, b and c) indicate significant differences among the same vegetation types in different soil layers (*P < 0.05*). SOC: soil organic carbon, DOC: dissolved organic carbon, MBC: microbial biomass carbon and EOC: easily oxidized organic carbon. SH: shrub, BF: bamboo forest, CFF: Chinese fir forest, MCBF: mixed conifer–broadleaf forest, and EBF: evergreen broad-leaved forest.

increasing soil depth (Table 6). Within the three soil layers, the L values for all the vegetation types were not significantly different, but the CPMI values were reversed. With regard to LI and CPI, significant difference was found only in the LI values of BF and CFF and in the CPI values of CFF and MCBF among the three soil layers. However, no significant difference in the average L values (0.17–0.28) was observed among these vegetation types. The mean LI value in BF was 0.65, which was significantly lower than those in the other vegetation types (0.97–1.08). Compared with the three other vegetation types, BF had significantly higher average CPI and CPMI values. In general, compared with that in SH, CPMI increased by 49.7%, 32.9% and 35.2% in BF, CFF, and EBF, respectively, whereas it decreased by 1.0 times in MCBF.

## Relationship between soil carbon fractions and soil physicochemical properties

As shown in Fig 3. the RDA analysis revealed that the first and second axes explained 98.89% and 0.74% of variability in soil carbon fractions, respectively, and the cumulative explanatory variance reached 99.63%. As observed, the two axes effectively illustrate the correlation between soil carbon fractions and its physicochemical factors, with the first axis serving as a primary determinant. The arrow lines of TP and TN were the longest, signifying that these two indicators had a greater impact on soil carbon fractions. All soil carbon fractions correlated positively with TN, TP, and moisture but negatively with BD. Notably, pH only exerted a negative effect on EOC and DOC.

## Discussion

### Effects of different vegetation restoration types on SOC concentrations and stocks

Vegetation restoration can enrich surface flora and fauna communities, increase soil inputs of organic matter (e.g., litter and animal residues), and promote SOC accumulation [35,36].

**Table 6. SOC pool indexes (L, LI, CPI, and CPMI) of the different vegetation types.**

| Soil layer | BF | CFF | MCBF | EBF |
|---|---|---|---|---|
| **L** | | | | |
| 0–10 cm | 0.16 ± 0.00 Ba | 0.28 ± 0.01 Aa | 0.29 ± 0.04 Aa | 0.28 ± 0.03 Aa |
| 10–20 cm | 0.18 ± 0.01 Ba | 0.32 ± 0.06 Aa | 0.28 ± 0.01 Aa | 0.27 ± 0.03 Aa |
| 20–30 cm | 0.17 ± 0.02 Aa | 0.25 ± 0.01 Aa | 0.20 ± 0.05 Aa | 0.26 ± 0.08 Aa |
| Average | 0.17 ± 0.01 A | 0.28 ± 0.03 A | 0.26 ± 0.04 A | 0.27 ± 0.05 A |
| **LI** | | | | |
| 0–10 cm | 0.47 ± 0.03 Bb | 0.85 ± 0.08 Ab | 0.87 ± 0.16 Aa | 0.85 ± 0.13 Aa |
| 10–20 cm | 0.76 ± 0.03 Ca | 1.35 ± 0.14 Aa | 1.17 ± 0.04 Ba | 1.11 ± 0.04 Ba |
| 20–30 cm | 0.72 ± 0.14 Aa | 1.05 ± 0.06 Ab | 0.88 ± 0.19 Aa | 1.12 ± 0.35 Aa |
| Average | 0.65 ± 0.07 B | 1.08 ± 0.10 A | 0.97 ± 0.13 A | 1.03 ± 0.18 A |
| **CPI** | | | | |
| 0–10 cm | 2.24 ± 0.45 Aa | 0.77 ± 0.14 Bb | 0.67 ± 0.15 Ba | 1.22 ± 0.13 Ba |
| 10–20 cm | 2.95 ± 0.57 Aa | 1.66 ± 0.36 Ba | 0.52 ± 0.11 Cab | 1.71 ± 0.38 Ba |
| 20–30 cm | 3.86 ± 0.85 Aa | 1.79 ± 0.40 Ba | 0.37 ± 0.05 Cb | 1.57 ± 0.34 Ba |
| Average | 3.02 ± 0.62 A | 1.40 ± 0.30 B | 0.52 ± 0.10 C | 1.50 ± 0.34 B |
| **CPMI** | | | | |
| 0–10 cm | 103.36 ± 13.34 Aa | 64.49 ± 4.78 Bb | 56.86 ± 6.27 Ba | 102.70 ± 11.40 Ab |
| 10–20 cm | 225.84 ± 54.10 Ab | 196.59 ± 46.38 Aa | 60.81 ± 4.40 Ba | 191.73 ± 49.81 Aa |
| 20–30 cm | 267.63 ± 20.92 Ab | 186.37 ± 31.43 Ba | 31.60 ± 8.76 Cb | 168.43 ± 15.56 Bab |
| Average | 198.94 ± 29.46 A | 149.15 ± 27.53 B | 49.76 ± 6.48 C | 154.29 ± 25.59 B |

L: Lability of C, LI: Lability index, CPI: Carbon pool index, and CPMI: Carbon pool management index. BF: bamboo forest, CFF: Chinese fir forest, MCBF: mixed conifer–broadleaf forest, and EBF: evergreen broad-leaved forest.

Various vegetation restoration practices exert significantly varying effects on SOC stocks due to differences in soil texture, aboveground biomass, litter quantity and quality, and root biomass. In the current study, the concentrations and stocks of SOC in five vegetation types demonstrated significantly higher values at the surface horizon (0–10 cm) compared with the underlying 10–20 cm and 20–30 cm layers. This finding could be attributed to the greater accumulation of litter debris in the topsoil layer, which facilitated the supply of nutrients, leading to higher microbial activity and population [10,37]. In addition, the soil gradually becomes more compact with depth, and this condition is not conducive to root penetration and the survival of soil microorganisms; therefore, SOC content gradually decreases [38].

At a depth of 0–30 cm, the mean concentration and total stocks of SOC were BF > EBF > CFF > SH > MCBF. These variations are primarily influenced by the amount, chemical composition, and decomposition rate of leaf litter, and root biomass. In BF, litterfall and abundant fine roots are beneficial for improvement of soil's fertility and SOC [39,40]. Although BF had the lowest amount of litterfall per unit area compared with SH, CFF, EBF, and MCBF, as measured in our previous research in this region [28], its litterfall decomposes faster than those of the others stands due to the abundance of the fungi *Mortierella*, which exhibit strong hydrolytic ability within BF, promoting litterfall decay [41,42]. Moreover, previous studies have confirmed that BF is rich in fine roots that can provide carbon and nutrients through rapid turnover and improve soil health [40]; furthermore, these roots demonstrate superior efficiency in binding soil particles to prevent soil erosion. In addition, BF had the lowest mean soil BD (1.48 ± 0.11 g·cm$^{-3}$) at a depth of 0–30 cm (Table 2). Lower BD indicates looser soil, higher soil porosity, and better air and water flow in the soil, which are beneficial for soil

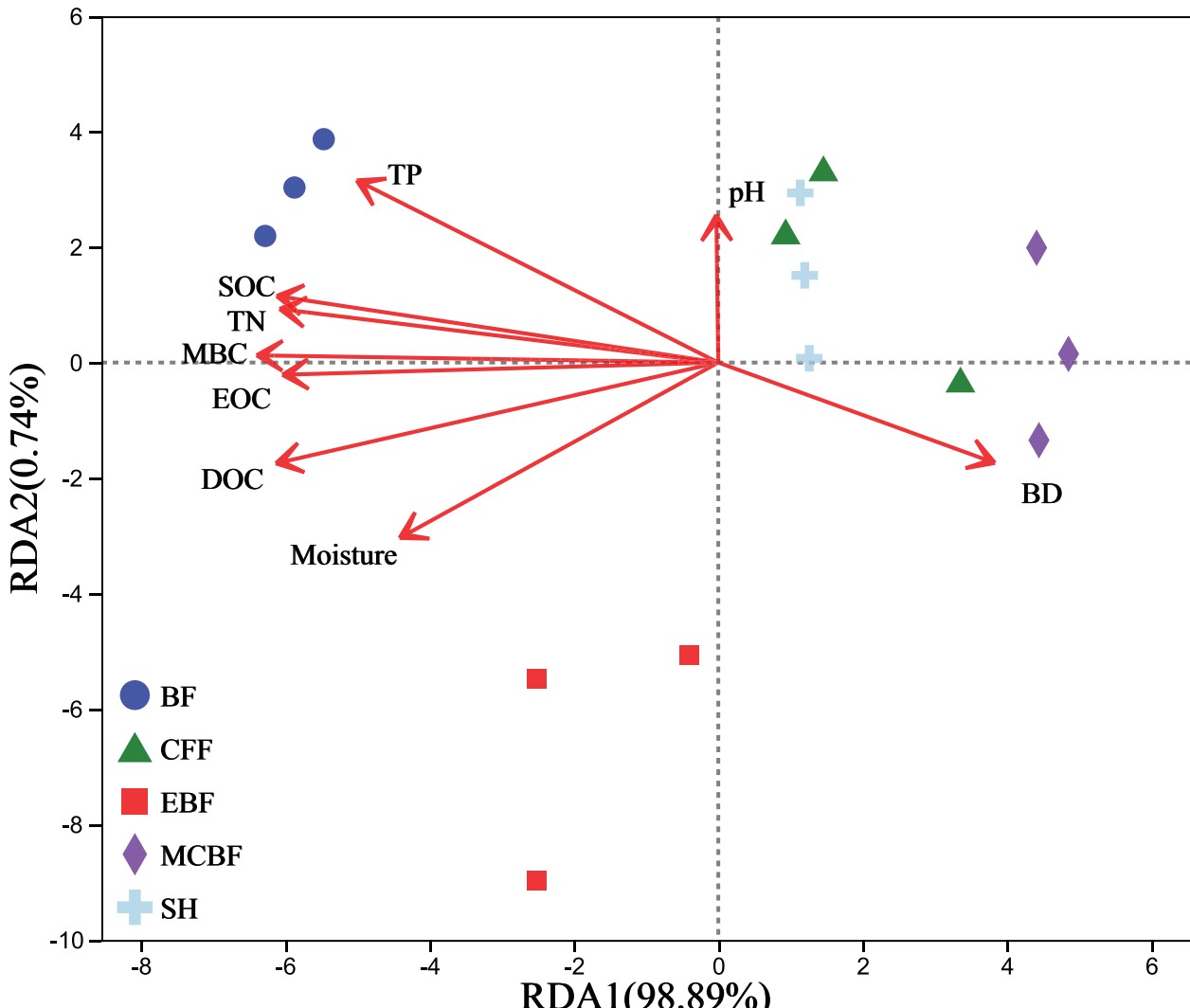

**Fig 3. Redundancy analysis of soil carbon fractions and physicochemical properties.** (Correlations among soil carbon fractions, soil physicochemical properties, and principal component analysis axes are represented by the length and angle of lines. The length of the arrow indicates the relative extent explained by that variables; **n** = 45. SOC: soil organic carbon, DOC: dissolved organic carbon, MBC: microbial biomass carbon and EOC: easily oxidized organic carbon. TN: total nitrogen, TP: total phosphorus, BD: bulk density, and pH: soil acid-alkalinity. SH: shrub, BF: bamboo forest, CFF: Chinese fir forest, MCBF: mixed conifer–broadleaf forest, and EBF: evergreen broad-leaved forest.).

microorganisms and root growth. Lower BD also promotes the production of root exudates, such as amino acids, which alter the C:N and pH by inhibiting microbial degradation, allowing SOC to accumulate [43]. Besides, the high elevation of BF (1028 m) as indicated in Table 1, contributed significantly to the accumulation of SOC. Garcia et al. (2016) [44] found that organic carbon and nitrogen were strongly and positively correlated with altitude. This conclusion was also confirmed in other studies, including Hou et al. (2019) [45], Zhang et al. (2021) [38] and Wang et al. (2023) [46], because lower soil moisture and temperature at higher altitudes result in a reduced decomposition rate of organic matter, augmenting the carbon sequestration capacity of soil [47,48]. Research has shown that the highly stable phytolith-occluded carbon in bamboo could potentially play a significant role in carbon storage and conservation [49,50]; this may be another reason for the highest SOC stocks in BF compared

with those in the other vegetation types. By comparison, MCBF had the lowest SOC concentration and stock; this finding was possibly related to the low coverage of coniferous litter contained therein and its difficulty to decompose [51]. Slope aspect also plays an important role in regulating carbon dynamics in some studies [52,53]. MCBF is located on the southeast slope, while the other vegetation types are observed on the northwest slope (Table 1). Soils on warmer, south-facing slopes tends to be highly mineralized, resulting in lower organic matter content [54]. Similar results were reported by Lozano-García et al.(2016) [44] and Bangroo et al.(2017)[32]. Compared with those in MCBF, the higher SOC concentrations and total stocks in EBF, CFF, and SH may be due to the relatively high litterfall biomass levels. EBF is characterized by large litter volume, high nutrient return, and rapid humification process. Therefore, it plays a crucial role in water conservation and biodiversity maintenance, surpassing that of coniferous forests. CFF had lower SOC concentration and total stocks than those of BF and EBF. The possible reason is that allelopathic substances, such as cyclic dipeptides, are released during the turnover of Chinese fir fine roots [55], which play a certain inhibitory role in the growth and reproduction of microorganisms. However, our results differ from those of previous studies conducted by some researchers who claimed that fir forests could increase SOC accumulation more than other plant species [38,56]. This discrepancy is attributed to the fact that many factors, such as climate, soil texture, plant residue amount, and topography, influence the effect of fir forests on SOC sequestration in different regions. Compared with MCBF, SH produces higher amounts of litter and roots, promoting SOC accumulation. Nevertheless, the better aeration and light transmittance of SH result in easier SOC decomposition than BF, EBF, and CFF. Therefore, management practices should focus on SH restoration to minimize $CO_2$ emission rates. However, except for BF, SOC concentrations in the other four vegetation types in surface soil layer (0-10 cm) were 10.05-18.36 $g·kg^{-1}$, lower than those of the Danxia natural forest in southeast China (24.32 $g·kg^{-1}$) [57]. Moreover, total SOC stocks in EBF, CFF, SH, and MCBF were 26.18-60.13 $t\ ha^{-1}$, lower than that in natural regeneration of secondary forest (74.07 $t\ ha^{-1}$) in Karst region of southwest China at a depth of 0-30 cm studied by Pang et al.[23].

## Changes in LOCFs and soil stocks

LOCFs play an important role in the improvement of soil quality due to their ability to maintain soil fertility and minimize negative environmental effects [58]. The relative proportions of different soil carbon fractions reflect the cycle and turnover rate of the soil carbon pool and determine soil quality and mineralization patterns [59,60]. Different vegetation restoration types significantly altered the quantity and quality of carbon source input, affecting the functional groups and abundance of microorganisms and the transformation process of soil matter. This condition, in turn, caused differences in stand LOCFs. For all the vegetation types, soil DOC, MBC (excluding BF and CFF), and EOC concentrations across the different vegetation types decreased with depth (Fig 2). Our results are essentially consistent with the studies on these soil quality indices by Xiao et al.[61] and de Moraes Sá et al.[62]. The topsoil layer typically boasts higher LOC contens compared to the deeper soil layers due to the large amount of litter return and root biomass accumulation. Meanwhile, the abundance of soil microorganisms at the soil surface drives the decomposition and turnover of soil organic matter [37].

Soil DOC is mostly derived from litterfall, root turnover, and exudation [63]. It is the primary energy source for microorganisms, affecting their composition and soil respiration rates [64,65]. Soil DOC exhibits certain characteristics, such as solubility, strong leaching, easy mobility, and short turnover time, and thus, it is easily lost [66]. This phenomenon is one of

the major mechanisms that contributes to SOC depletion. Filep et al. [67] demonstrated the high potential of DOC in indicating soil quality and fertility. Thus, the higher DOC levels in BF and EBF could denote overall better soil condition. The highly heterogeneous nature of aboveground vegetation composition in forest ecosystems should induce high DOC variability in forest soils as observed in the current study. Soil DOC:SOC largely reflects the stability and loss of SOC. A higher ratio infers greater activity of SOC, and consequently, its worse stability [68]. In general, soil DOC content does not exceed 2% of SOC [69]. DOC accounted for a small proportion of SOC (0.39%–1.64%) in our study (Table 2), which was significantly lower than that in karst soils (2.37%–8.85%), as reported by Pang et al. [23]. The higher average DOC:SOC in MCBF might be related to its lowest SOC content.

Soil MBC can rapidly reflect changes in soil organic matter and nutrients; it is an effective index for evaluating the dynamics and quality of SOC [70]. In our study, the total MBC stocks of the five vegetation types at 0–30 cm soil depth varied from 0.50 t·ha$^{-1}$ to 0.96 t·ha$^{-1}$, which were lower than those of the secondary forest, Eucalyptus maiden, and *Pinus yunnanensis* in the karst forest of southwest China studied by Pang et al. [23] (1.34–2.48 t·ha$^{-1}$). Chen et al. [71] also confirmed that MBC contents were greater within karst forests compared to that in the other areas. The concentrations and stocks of MBC in BF and EBF were higher than those in SH, CFF, and MCBF (Table 3). This is most likely due to the abundance of easily decomposable litter in these two stands providing rich nutrients for soil microbes. The variation in MBC:SOC ratios can effectively evaluate the short-term effects of forest management measures and land use alterations [72]. In general, the higher MBC:SOC, the more soil carbon pool accumulates, the greater the utilization efficiency of soil organic matter by microorganisms, and the better the soil quality [73]. For the five vegetation types, MBC held 0.86%–3.19% of SOC, which was basically within the range (1–5%) defined by previous studies [74]. Higher MBC:SOC was found in MCBF compared with those of the four other vegetation types, indicating that soil organic matter was converted into microbial biomass more efficiently in MCBF.

EOC is typified by a swift turn around time, and the proportion of EOC in SOC is relatively large; hence, changes in soil carbon pool capacity mostly occurs in EOC [75]. Our study showed that EOC concentrations and stocks were higher in BF and EBF soils compared with soil in SH, CFF, and MCBF. According to Blair et al[19], EOC typically comprises 5–30% of SOC. In our study, however, EOC:SOC was 13.45–26.51% in five vegetation types. A bigger proportion of EOC to SOC will indicate a higher rate of nutrient decomposition, leading to less buildup of soil carbon, and vice versa [14]. However, the lowest mean EOC:SOC (14.35%) was recorded in BF at a depth of 0–30 cm, demonstrating that BF was a highly effective species for soil carbon sequestration due to its vigorous growth rate.

## SOC pool indexes change among the different vegetation types

CPMI is an important parameter for evaluating SOC dynamics in the process of soil environment change; it can reflect the influence of different degrees of external interference on soil quality [76]. A larger L value indicates that SOC is more susceptible to decomposition by microorganisms, and thus, is absorbed and utilized by plants [77]. No significant difference in mean L value among BF, CFF, EBF, and MCBF was observed in our study. This phenomenon suggests that the physical protection provided by soil organic matter is good and the proportion of carbon lability in soil is similar in the four vegetation types. The LI value of BF was significantly lower than those of the three other vegetation types, indicating that SOC in BF exhibited better stability and slower transformation [23]. CPI predominantly reflects the changes of SOC, while CPMI is serves to monitor soil carbon restoration and degradation.

Therefore, greater CPMI values signify the rejuvenation of soil carbon, indicating a healthier ecosystem; conversely, smaller CPMI values mirror the system's ongoing degradation, necessitating intervention measures [10,19]. In the five vegetation types, the average values of CPI and CPMI exhibited the trend of BF> EBF> CFF> MCBF. The findings prompt us to speculate that soil in BF might potentially accumulate more SOC and LOCFs than soils in other vegetation types. Meanwhile, MCBF had the lowest CPI and CPMI among all the vegetation types, again reflecting poor carbon input as evidence from the lower fine root biomass and difficult-to-decompose litterfall.

## Effects of soil physicochemical properties on SOC and its labile fractions

Vegetation restoration projects have increased plant diversity and soil nutrients, altering soil physicochemical properties and further affecting soil carbon stocks (Fig 3). Soil BD is a sensitive indicator of soil compaction, and its value is closely related to SOC content. A negative linear association was discerned between soil carbon fractions and BD. In general, the higher the soil BD, the smaller the soil porosity, and the worse the air permeability, inhibiting soil biological activity, root growth, and development, and then leading to soil carbon content reduction. In addition to BD, pH also plays a key role in regulating SOC content. Soil pH exerted negative effects on EOC and DOC in the present study. This finding is consistent with the existing research conclusions of [78]. Soil pH directly affects soil microbial quantity and activity, regulating the soil carbon conversion process. Only SH soil is alkaline, while the other vegetation soils are acidic (Table 2). An acidic environment is not conducive to the dissolution of DOC, and an increase in pH helps improve the dissolution of DOC [79]. However, an excessively high or low pH can exert a significant effect on the soil carbon levels, largely because soil pH directly influences microbial number and activity. The pH of bacteria that determines the decomposition of organic carbon sources is neutral and alkaline. Nitrogen and phosphorus are essential nutrients for plant growth, and increasing their levels can promote the accumulation of SOC [80]. TN and TP have notably positive association with SOC and LOCFs, as revealed by this study. The mass fraction of SOC depends on the amount of nitrogen content in soil to a certain extent [81], because TN can promote plant growth, increase litter quantity and quality, and enhance organic matter inputs. Furthermore, soil TN content also influences decomposition rate of organic matter, which is important for the cycling of nutrients in soil [78]. Some studies have also reported that SOC is positively correlated with soil TP [82]. Phosphorus in soil affects SOC content by increasing the input of synthetic organic matter substrates and the activity of soil microorganisms [83]. Wang et al. suggested that strong coupling between soil nutrients (i.e., TN and TP) and microbial biomass (i.e., MBC, microbial biomass nitrogen, and microbial biomass phosphorus) during vegetation restoration [16]. Therefore, strengthening soil nitrogen and phosphorus management in the soil degradation area of China Danxia may be more beneficial for ecosystem restoration. Overall, soil carbon fractions will be directly influenced by soil physicochemical properties following revegetation. Different durations, degrees of cover, and types of vegetation restoration have different effects on soil factors, and the effect of soil restoration on carbon fractions varies accordingly.

## Conclusions

Our results indicated that SOC, LOCFs and CPMI were significantly affected by different vegetation restoration types in Chishui Danxia district. The contents and stocks of SOC, DOC, MBC (except for BF and CFF), and EOC in all the vegetation types decreased with soil depth. BF, EBF, and CFF resulted in higher SOC concentration and stocks, and CPMI than SH and

MCBF. Moreover, BF and EBF exhibited significantly higher soil concentrations of DOC, MBC, and EOC, and total stocks of MBC and EOC, compared with the three other vegetation types. Therefore, BF and EBF can be more ideal vegetation types for rapidly restoring degraded soil and promoting ecological services in the Danxia regions of southwest China. The accumulation of soil carbon fractions would be strongly influenced by TN, TP, and moisture in Danxia areas as vegetation restoration progressed. According to our findings, CPMI is a comprehensive assessment index of soil quality that can reflect the extent to which soil quality is degraded or renewed by different management practices in the ecologically fragile Danxia landform area.

## Supporting information

**S1 Text.** S1 Table. Mean (±SE, n = 3) soil basic physical and chemical properties in the five vegetation types. S2 Table.Two-way ANOVA of the effects of different vegetation types (VT) and soil depths (SD) on SOC and LOCFs. S3 Table.The proportion of LOCFs (DOC, MBC, and EOC) in total SOC (%) of the five vegetation types. S4 Table. LOCFs (DOC, MBC, and EOC) stocks in the different vegetation types. S5 Table. SOC pool indexes (L, LI, CPI, and CPMI) of the different vegetation types. S1 Fig. SOC concentration (a) and stocks (b) in each soil layer of the different vegetation types. S2 Fig. LOCF concentrations of the different vegetation types. a) DOC, b) MBC, and c) EOC. S3 Fig. Redundancy analysis of soil carbon fractions and physicochemical properties. (ZIP)

## Acknowlegements

We are especially grateful to the staff of Xishui Subtropical Evergreen Broad-leaved Forest National Nature Reserve for their strong support and assistance in the field investigation and sampling during the pre-study period.

## Author contributions

**Conceptualization:** Ye Xiao, Yuguang Zhang.

**Data curation:** Ye Xiao, Yuguang Zhang, Zhigang Huang.

**Formal analysis:** Ye Xiao, Yuguang Zhang, Jiumei Long, Kang Luo, Zhigang Huang.

**Funding acquisition:** Zhigang Huang.

**Investigation:** Ye Xiao, Jiumei Long, Kang Luo, Zhigang Huang.

**Methodology:** Ye Xiao, Yuguang Zhang, Zhigang Huang.

**Project administration:** Zhigang Huang.

**Resources:** Ye Xiao, Zhigang Huang.

**Software:** Yuguang Zhang.

**Supervision:** Ye Xiao, Zhigang Huang.

**Validation:** Zhigang Huang.

**Visualization:** Yuguang Zhang.

**Writing – original draft:** Ye Xiao, Zhigang Huang.

**Writing – review & editing:** Ye Xiao.

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
