## [Decision Letter · Decision Letter 0]

14 Oct 2024

PONE-D-24-34559Responses of soil labile organic carbon stocks and carbon pool management index to different vegetation restoration in Danxia landform region of southwest ChinaPLOS ONE

Dear Dr. Huang,

Thank you for submitting your manuscript. After a thorough review, we have received feedback from the reviewers, and we kindly ask you to address the comments and suggestions they have provided.

To proceed with the revision, please carefully consider each point raised by the reviewers. Your revised manuscript should reflect these changes to improve clarity, scientific accuracy, and overall quality.

We look forward to receiving your revised manuscript.

Kind regards,

Taimoor Hassan Farooq

Academic Editor

PLOS ONE

Journal Requirements:

1. When submitting your revision, we need you to address these additional requirements. Please ensure that your manuscript meets PLOS ONE's style requirements, including those for file naming. The PLOS ONE style templates can be found at https://journals.plos.org/plosone/s/file?id=wjVg/PLOSOne_formatting_sample_main_body.pdf and https://journals.plos.org/plosone/s/file?id=ba62/PLOSOne_formatting_sample_title_authors_affiliations.pdf 2. We noticed you have some minor occurrence of overlapping text with the following previous publication(s), which needs to be addressed: Responses of soil labile organic carbon fractions and stocks to different vegetation restoration strategies in degraded karst ecosystems of southwest China - https://doi.org/10.1016/j.ecoleng.2019.08.008 Build-up of labile, non-labile carbon fractions under fourteen-year-old bamboo plantations in the Himalayan foothills -https://doi.org/10.1016/j.heliyon.2021.e07850 (Among others) In your revision ensure you cite all your sources (including your own works), and quote or rephrase any duplicated text outside the methods section. Further consideration is dependent on these concerns being addressed. 3. In your Methods section, please provide additional information regarding the permits you obtained for the work. Please ensure you have included the full name of the authority that approved the field site access and, if no permits were required, a brief statement explaining why. 4. Thank you for stating the following financial disclosure: "Guizhou Provincial Basic Research Program (Natural Science) (Qiankehejichu-ZK[2024]yiban 674 and 691) and Scientific Research Project of Zunyi Normal University (Zunshi BS [2019]30)." Please state what role the funders took in the study.  If the funders had no role, please state: "The funders had no role in study design, data collection and analysis, decision to publish, or preparation of the manuscript." If this statement is not correct you must amend it as needed. Please include this amended Role of Funder statement in your cover letter; we will change the online submission form on your behalf. 5. Thank you for stating the following in the Acknowledgments Section of your manuscript: "We gratefully acknowledge the Supported by Guizhou Provincial Basic Research Program (Natural Science) (Qiankehejichu-ZK[2024]yiban 674 and 691) and Scientific Research Project of Zunyi Normal University (Zunshi BS [2019]30)." We note that you have provided funding information that is not currently declared in your Funding Statement. However, funding information should not appear in the Acknowledgments section or other areas of your manuscript. We will only publish funding information present in the Funding Statement section of the online submission form. Please remove any funding-related text from the manuscript and let us know how you would like to update your Funding Statement. Currently, your Funding Statement reads as follows: "Guizhou Provincial Basic Research Program (Natural Science) (Qiankehejichu-ZK[2024]yiban 674 and 691) and Scientific Research Project of Zunyi Normal University (Zunshi BS [2019]30)." Please include your amended statements within your cover letter; we will change the online submission form on your behalf. 6. We note that your Data Availability Statement is currently as follows: "All relevant data are within the manuscript and its Supporting Information files." Please confirm at this time whether or not your submission contains all raw data required to replicate the results of your study. Authors must share the “minimal data set” for their submission. PLOS defines the minimal data set to consist of the data required to replicate all study findings reported in the article, as well as related metadata and methods (https://journals.plos.org/plosone/s/data-availability#loc-minimal-data-set-definition). For example, authors should submit the following data: - The values behind the means, standard deviations and other measures reported;- The values used to build graphs;- The points extracted from images for analysis. Authors do not need to submit their entire data set if only a portion of the data was used in the reported study. If your submission does not contain these data, please either upload them as Supporting Information files or deposit them to a stable, public repository and provide us with the relevant URLs, DOIs, or accession numbers. For a list of recommended repositories, please see https://journals.plos.org/plosone/s/recommended-repositories. If there are ethical or legal restrictions on sharing a de-identified data set, please explain them in detail (e.g., data contain potentially sensitive information, data are owned by a third-party organization, etc.) and who has imposed them (e.g., an ethics committee). Please also provide contact information for a data access committee, ethics committee, or other institutional body to which data requests may be sent. If data are owned by a third party, please indicate how others may request data access. 7. PLOS requires an ORCID iD for the corresponding author in Editorial Manager on papers submitted after December 6th, 2016. Please ensure that you have an ORCID iD and that it is validated in Editorial Manager. To do this, go to ‘Update my Information’ (in the upper left-hand corner of the main menu), and click on the Fetch/Validate link next to the ORCID field. This will take you to the ORCID site and allow you to create a new iD or authenticate a pre-existing iD in Editorial Manager. 8. Please amend either the title on the online submission form (via Edit Submission) or the title in the manuscript so that they are identical. 9. Please include captions for your Supporting Information files at the end of your manuscript, and update any in-text citations to match accordingly. Please see our Supporting Information guidelines for more information: http://journals.plos.org/plosone/s/supporting-information.

Additional Editor Comments:

Dear authors,

Thank you for submitting your manuscript. After a thorough review, we have received feedback from the reviewers, and we kindly ask you to address the comments and suggestions they have provided.

To proceed with the revision, please carefully consider each point raised by the reviewers. Your revised manuscript should reflect these changes to improve clarity, scientific accuracy, and overall quality.

Reviewers' comments:

Reviewer's Responses to Questions

**Comments to the Author**

1. Is the manuscript technically sound, and do the data support the conclusions?

Reviewer #1: Yes

Reviewer #2: Yes

2. Has the statistical analysis been performed appropriately and rigorously? 

Reviewer #1: Yes

Reviewer #2: Yes

3. Have the authors made all data underlying the findings in their manuscript fully available?

Reviewer #1: Yes

Reviewer #2: No

4. Is the manuscript presented in an intelligible fashion and written in standard English?

Reviewer #1: Yes

Reviewer #2: Yes

5. Review Comments to the Author

Reviewer #1: Manuscript Number: PONE-D-24-34559

Title: Responses of soil labile organic carbon stocks and carbon pool management index to different vegetation restoration in Danxia landform region of southwest China

This study aimed to investigate how different vegetation restoration types, including shrub and bamboo, fir, evergreen broad-leaved, mixed coniferous-broadleaf forests, influence the stocks of soil labile organic C and C pool management index in the Danxia landform of Southwest China. The research subject may demonstrate good outcomes of scientific research on rehabilitating degraded lands and ecological restoration. Overall, this manuscript was well written, but minor revisions are needed to improve in some parts, including English errors, before publication in this journal.

Specific comments

1. Abstract

- L15: Delete ‘in China’s Danxia region’

- L34: exhibit?

2. Introduction

- L77: Blair and Lefroy [20]

3. Materials and methods

- L153: by Ghani et al. [29].

4. Result

- L259: Check the use of abbreviations. Does lability (L) indicate ‘Lability of C’ shown in L176’?

5. Discussion

- L309: Table 2

- L361: Filep et al. [66]

- L375: by Pang et al. [24]; Chen et al. [70]

6. Tables & Figures

- Table and figure legends should be improved, including statistic description.

7. Reference list

- It should be revised. Please check the submission guidelines of the PLOS ONE journal.

Reviewer #2: Based on my review of the manuscript entitled "Responses of soil labile organic carbon stocks and carbon pool management index to different vegetation restoration in Danxia landform region of southwest China", I hereby forward major revisions. The manuscript highlights the importance of SOC as an index for evaluating soil quality and outlines the main vegetation types examined, including shrub, bamboo forest, Chinese fir forest, evergreen broad-leaved forest, and mixed coniferous-broadleaf forest. The key results regarding the impact of these vegetation types on SOC and labile organic carbon fractions (LOCFs) like dissolved organic carbon (DOC), microbial biomass organic carbon (MBC), and easily oxidizable organic carbon (EOC) are well-presented. The manuscript effectively sets the stage for understanding the ecological significance of the findings.

The introduction could be improved by a more detailed discussion on the gaps in current research specifically relating to SOC dynamics in the Danxia landform under different vegetation restorations. References to previous studies are appropriately included, yet a clearer linkage between these studies and the current research question would enhance the manuscript's introduction.

The discussion would benefit from a more detailed comparison with other similar ecosystems globally, which would provide a broader context for the findings.

Other comments:

- Statistical analysis: Did you check residual normality? Please explain in the manuscript.

- Figures and Tables: Clarify which letters are capitalized and which are lowercase, and specify whether the error measurement is standard deviation (SD) or standard error (SE). Please provide the full forms of all abbreviations used. Each figure and table should be self-explanatory, ensuring that the reader can understand the content without referring to the main text.

- Redundancy analysis: Please specify which software was used to perform the RDA in your methods section. Providing the exact software and version used will help readers replicate your results. Additionally, make sure to mention any specific settings, packages, or libraries applied during the analysis. The abbreviations such as "SOC", "TN", "MBC", "EOC", "DOC", "BD", "TP", and "pH" are not explained within the figure. For clarity, each abbreviation should be fully defined in either the caption or the legend. "RDA1 (98.89%)" and "RDA2 (0.74%)" show the percentage of the variance explained by the respective axes, which is good, but a clearer explanation of what these represent in the context of the study might be needed in the caption.

6. PLOS authors have the option to publish the peer review history of their article (what does this mean? ). If published, this will include your full peer review and any attached files.

**Do you want your identity to be public for this peer review?** For information about this choice, including consent withdrawal, please see our Privacy Policy .

Reviewer #1: **Yes: ** Young-Nam Kim

Reviewer #2: **Yes: ** Ali Mokhtassi-Bidgoli

---

## [Author Response · Author response to Decision Letter 0]

27 Nov 2024

Dear Reviewers and Editors :

Thank you very much for your constructive comments and valuable recommendations concerning our manuscript entitled “Responses of soil labile organic carbon stocks and the carbon pool management index to different vegetation restoration types in the Danxia landform region of southwest China” (manuscript number: PONE-D-24-34559). These comments are very helpful for revising and improving our paper, as well as the important guiding significance to our researches. We have carefully revised the manuscript according to your suggestions which we hope meet with approval. Revised portion are marked in red in the revised manuscript (Revised manuscript with Track changes). Our response to the comments are listed below:

Response to reviewer 1:

Comment 1: Abstract

- L15: Delete ‘in China’s Danxia region’

- L34: exhibit?

Reply:Thank you very much for the advice. We have deleted ‘in China’s Danxia region’ in line 15 and correct the incorrect word "exbibit" to"exhibited" in line 34 in our revised manuscript, please see our revised manuscript page1, lines 15; page2, 33.

Comment 2: Introduction

- L77: Blair and Lefroy [20]

Reply:Thank you very much. We have amended "Blair and Lefroy" to" Blair et al.", please see the introduction in our revised manuscript page4, lines 74.

Comment 3: Materials and methods

- L153: by Ghani et al. [29].

Reply:Thank you very much. We have added “Ghani et al.” in L157, please see the ‘Materials and methods’ section in our revised manuscript page 8, lines 157.

Comment 4: Result

- L259: Check the use of abbreviations. Does lability (L) indicate ‘Lability of C’ shown in L176’?

Reply: We scrutinized the use of abbreviations. Yes, the lability (L) in L259 indicate ‘Lability of C’ shown in L176. To make it clearer to readers, we have added the abbreviation ‘(L)’ to formula (3) and changed ‘lability (L)’ in L259 to ‘lability of C (L)’. The following is the revised formula(3) :

Please see the ‘Result’ section in our revised manuscript page 10, lines 179 and page 15, lines 281.

Comment 5: Discussion

- L309: Table 2

- L361: Filep et al. [66]

- L375: by Pang et al. [24]; Chen et al. [70]

Reply:Thank you very much. We have corrected ‘Table 1’to ‘Table 2’ in L309. Filep et al.(2015) [66] in L361 and by [24], Chen et al. (2019)[70] in L375 also were modified into ‘Filep et al. [66]’ in L361 and ‘by Pang et al. [24]; Chen et al. [70]’ in L375. Please see the ‘Discussion’ section in our revised manuscript page 18,line 340; page 21, line 405 and page 21, line 411.

Comment 6: Tables & Figures

- Table and figure legends should be improved, including statistic description.

Reply: We have modified the tables and figures to make it clear which letters are capitalized and which are lowercase, and to specify clearly that the error measurement is standard error (SE). The statistical description is clearer. In addition, each abbreviation is fully defined in each table and figure legends. Please see the tables and figures in our revised manuscript.

Comment 7:Reference list

- It should be revised. Please check the submission guidelines of the PLOS ONE journal.

Reply:We have revised the references format according to the submission guidelines of the PLOS ONE journal, please see the revised Reference section, page 27-32, line 533-767.

Response to reviewer #2:

Comment 1:The introduction could be improved by a more detailed discussion on the gaps in current research specifically relating to SOC dynamics in the Danxia landform under different vegetation restorations. References to previous studies are appropriately included, yet a clearer linkage between these studies and the current research question would enhance the manuscript's introduction.

Reply: We added the sentence ‘As for the Danxia landform area, people’s research mainly focuses on the vegetation, biodiversity, geological structure and tourism development [26,27,3]’ in our revised manuscript, page 5, line 99-101, to illustrate the previous research on Danxia and highlight the signifiance of our research.

The refenrences were added:

[26]Yang J, Xiong K, Xiao S. Spatial temporal variation of vegetation coverage of reddish landform in Chishui Cityi.Yangtze River, 2018, 49 (18): 38-44. https://doi.org/10.16232/j.cnki.1001-4179.2018.18.009.

[27]Meng, S. Y., Chen, B., & Sodmergen. (2023). Sedum danxiacola (Crassulaceae), an endemic new species from the Danxia landform, China. Kew Bulletin, 78(4), 673-681. https://doi.org/10.1007/S12225-023-10122-Y.

Comment 2:The discussion would benefit from a more detailed comparison with other similar ecosystems globally, which would provide a broader context for the findings.

Reply:In the discussion section, we added the following:

However, except for BF, SOC concentrations in the other four vegetation types in surface soil layer (0-10 cm) were 10.05-18.36 g·kg-1, lower than those of the Danxia natural forest in southeast China (24.32 g·kg-1) [58]. Moreover, total SOC stocks in EBF, CFF, SH, and MCBF were 26.18-60.13 t ha−1, lower than that in natural regeneration of secondary forest (74.07 t ha−1) in Karst region of southwest China at a depth of 0-30 cm studied by Pang et al. (2019)[23].

Please see the revised manuscript, page 20, line 373-378.

Comment 3: Statistical analysis: Did you check residual normality? Please explain in the manuscript.

Reply: We have checked residual normality of data in the statistical analysis, which we have explained in our revised manuscript, please see page 10,183-184.

Comment 4: Figures and Tables: Clarify which letters are capitalized and which are lowercase, and specify whether the error measurement is standard deviation (SD) or standard error (SE). Please provide the full forms of all abbreviations used. Each figure and table should be self-explanatory, ensuring that the reader can understand the content without referring to the main text.

Reply: We have modified the tables and figures to make it clear which letters are capitalized and which are lowercase, and to specify clearly that the error measurement is standard error (SE). The statistical description is clearer. In addition, each abbreviation is fully defined in each table and figure legends. Please see the tables and figures in our revised manuscript.

Comment 5: Redundancy analysis: Please specify which software was used to perform the RDA in your methods section. Providing the exact software and version used will help readers replicate your results. Additionally, make sure to mention any specific settings, packages, or libraries applied during the analysis. The abbreviations such as "SOC", "TN", "MBC", "EOC", "DOC", "BD", "TP", and "pH" are not explained within the figure. For clarity, each abbreviation should be fully defined in either the caption or the legend. "RDA1 (98.89%)" and "RDA2 (0.74%)" show the percentage of the variance explained by the respective axes, which is good, but a clearer explanation of what these represent in the context of the study might be needed in the caption.

Reply: We have clarified in the statistical analysis section that the Redundancy Analysis (RDA) is carried out on the online tool of Majorbio Cloud Platform (https://cloud. Majorbio.com/page/tools/), please see the revised manuscript page 10, line 190-191. The all abbreviations in the figure (RDA) also are explained in figure legend, please see the revised manuscript page 17, line 311-314. For "RDA1 (98.89%)" and "RDA2 (0.74%)" showing the percentage of variance interpreted by the respective axes, we added an explanation“As observed, the two axes effectively illustrate the correlation between soil C fractions and its physicochemical properties, with the first axis serving as a primary determinant”, see the revised version, please see the the revised manuscript page 16, line 300-301.

Response to the editor’s comments:

1.Please ensure that your manuscript meets PLOS ONE's style requirements, including those for file naming. The PLOS ONE style templates can be found at：https://journals.plos.org/plosone/s/file?id=wjVg/PLOSOne_formatting_sample_main_body.pdf and 

Reply: We have modified the format according to the PLOS ONE style. Please see the revised manuscript.

2.We noticed you have some minor occurrence of overlapping text with the following previous publication(s), which needs to be addressed:

Responses of soil labile organic carbon fractions and stocks to different vegetation restoration strategies in degraded karst ecosystems of southwest China - https://doi.org/10.1016/j.ecoleng.2019.08.008

Build-up of labile, non-labile carbon fractions under fourteen-year-old bamboo plantations in the Himalayan foothills -https://doi.org/10.1016/j.heliyon.2021.e07850

(Among others)

In your revision ensure you cite all your sources (including your own works), and quote or rephrase any duplicated text outside the methods section. Further consideration is dependent on these concerns being addressed.

Reply: We have revised and restated the duplicates you mentioned, and checked all cited sources (including our own works). Please see the revised manuscript.

3.In your Methods section, please provide additional information regarding the permits you obtained for the work. Please ensure you have included the full name of the authority that approved the field site access and, if no permits were required, a brief statement explaining why.

Reply: We added “Our research obtained the forestry administrative license of Xishui National Nature Reserve Administration of Guizhou Province.”Please see the “Sample collection and chemical analyses” in our Methods section, page 7, line 128-129.

We have included the full name of the authority that approved the field site access.

4.Please state what role the funders took in the study. If the funders had no role, please state: "The funders had no role in study design, data collection and analysis, decision to publish, or preparation of the manuscript. If this statement is not correct you must amend it as needed.

Reply: Our funding is funded by government agencies. Funding agencies provide financial support for our research design, data collection and analysis, publication decisions or manuscript preparation.

5.We note that you have provided funding information that is not currently declared in your Funding Statement. However, funding information should not appear in the Acknowledgments section or other areas of your manuscript. We will only publish funding information present in the Funding Statement section of the online submission form. Please remove any funding-related text from the manuscript and let us know how you would like to update your Funding Statement. Currently, your Funding Statement reads as follows: "Guizhou Provincial Basic Research Program (Natural Science) (Qiankehejichu-ZK[2024]yiban 674 and 691) and Scientific Research Project of Zunyi Normal University (Zunshi BS [2019]30)."

Reply: We have already deleted the funding information in the Acknowledgments section. We have provided a funding statement in cover letter; we wish editor change the online submission form on our behalf, our Funding Statement as follows: "Guizhou Provincial Basic Research Program (Natural Science) (Qiankehejichu-ZK[2024]yiban 674 and 691) and Scientific Research Project of Zunyi Normal University (Zunshi BS [2019]30)." .

6.Please confirm at this time whether or not your submission contains all raw data required to replicate the results of your study. Authors must share the “minimal data set” for their submission. PLOS defines the minimal data set to consist of the data required to replicate all study findings reported in the article, as well as related metadata and methods.

Reply: We confirm at this time our submission contains all raw data required to replicate the results of our study. The raw data please see the “Supporting information”.

7.If there are ethical or legal restrictions on sharing a de-identified data set, please explain them in detail (e.g., data contain potentially sensitive information, data are owned by a third-party organization, etc.) and who has imposed them (e.g., an ethics committee). Please also provide contact information for a data access committee, ethics committee, or other institutional body to which data requests may be sent. If data are owned by a third party, please indicate how others may request data access.

Reply: There are no ethical or legal restrictions on the data we submit.

8.PLOS requires an ORCID iD for the corresponding author in Editorial Manager on papers submitted after December 6th, 2016. Please ensure that you have an ORCID iD and that it is validated in Editorial Manager. To do this, go to ‘Update my Information’ (in the upper left-hand corner of the main menu), and click on the Fetch/Validate link next to the ORCID field. This will take you to the ORCID site and allow you to create a new iD or authenticate a pre-existing iD in Editorial Manager.

Reply: The corresponding author has registered an ORCID iD (0009-0006-0617-2462).

9.Please amend either the title on the online submission form (via Edit Submission) or the title in the manuscript so that they are identical.

Reply: We checked that the title in the manuscript was consistent with the title submitted online.

10.Please include captions for your Supporting Information files at the end of your manuscript, and update any in-text citations to match accordingly. Please see our Supporting Information guidelines for more information:

Reply: We include captions for our Supporting Information files at the end of our manuscript, and update any in-text citations to match accordingly. Please see the revised manuscript.

Special thanks to reviewers for your good comments and suggestions. We have tried our best to improve the manuscript, and here we did not list the changes but marked in red in revised paper. We appreciate for Reviewers and Editors warm work earnestly, and hope that the correction will meet with approval.

---

## [Decision Letter · Decision Letter 1]

12 Jan 2025

Responses of soil labile organic carbon stocks and the carbon pool management index to different vegetation restoration types in the Danxia landform region of southwest China

PONE-D-24-34559R1

Dear Dr. Huang,

We’re pleased to inform you that your manuscript has been judged scientifically suitable for publication and will be formally accepted for publication once it meets all outstanding technical requirements.

Kind regards,

Taimoor Hassan Farooq

Academic Editor

PLOS ONE

Additional Editor Comments (optional):

Dear authors,

I am pleased to inform you that your manuscript has been accepted for publication in PLOS One. After careful consideration and review, we are confident that your work will be a valuable contribution to the field.

Please ensure that all necessary revisions and final formatting are completed in accordance with the journal's guidelines.

Reviewers' comments:

Reviewer's Responses to Questions

**Comments to the Author**

1. If the authors have adequately addressed your comments raised in a previous round of review and you feel that this manuscript is now acceptable for publication, you may indicate that here to bypass the “Comments to the Author” section, enter your conflict of interest statement in the “Confidential to Editor” section, and submit your "Accept" recommendation.

Reviewer #1: All comments have been addressed

Reviewer #2: All comments have been addressed

2. Is the manuscript technically sound, and do the data support the conclusions?

Reviewer #1: Yes

Reviewer #2: (No Response)

3. Has the statistical analysis been performed appropriately and rigorously? 

Reviewer #1: Yes

Reviewer #2: (No Response)

4. Have the authors made all data underlying the findings in their manuscript fully available?

Reviewer #1: Yes

Reviewer #2: (No Response)

5. Is the manuscript presented in an intelligible fashion and written in standard English?

Reviewer #1: Yes

Reviewer #2: (No Response)

6. Review Comments to the Author

Reviewer #1: The revised manuscript has been improved overall by responding well to the reviewers' comments. In addition, the entire text, including the description of the experimental method (statistics, etc.), has been appropriately revised and appears to be well written by authors.

Reviewer #2: (No Response)

7. PLOS authors have the option to publish the peer review history of their article (what does this mean? ). If published, this will include your full peer review and any attached files.

**Do you want your identity to be public for this peer review?** For information about this choice, including consent withdrawal, please see our Privacy Policy .

Reviewer #1: No

Reviewer #2: **Yes: ** Ali Mokhtassi-Bidgoli

---

## [Editor Report · Acceptance letter]

PONE-D-24-34559R1

PLOS ONE

Dear Dr. Huang,

I'm pleased to inform you that your manuscript has been deemed suitable for publication in PLOS ONE. Congratulations! Your manuscript is now being handed over to our production team.

Kind regards,

on behalf of

Taimoor Hassan Farooq

Academic Editor

PLOS ONE